# The Site of Origin of Canine Abdominal Masses Correlates with the Risk of Malignancy: Retrospective Study of 123 Cases

**DOI:** 10.3390/ani11040962

**Published:** 2021-03-30

**Authors:** Paola Valenti, Valeria Pellegrino, Luisa Vera Muscatello, Barbara Brunetti, Elisa Zambon, Gian Marco Gerboni, Monica Alberti, Giancarlo Avallone

**Affiliations:** 1Clinica Veterinaria Malpensa, Samarate, 21017 Varese, Italy; pvalenti.dvm@gmail.com (P.V.); gerbonigianmarco@yahoo.it (G.M.G.); 2Department of Veterinary Medical Sciences, University of Bologna, Ozzano dell’Emilia, 40064 Bologna, Italy; valeria.pellegrino3@unibo.it (V.P.); b.brunetti@unibo.it (B.B.); giancarlo.avallone@unibo.it (G.A.); 3Ospedale Veterinario I Portoni Rossi, Zola Predosa, 40069 Bologna, Italy; elisa.zambon@portonirossi.it (E.Z.); Monica.Alberti@portonirossi.it (M.A.)

**Keywords:** mass, abdomen, canine, diagnosis, malignancy, histopathology

## Abstract

**Simple Summary:**

The detection of abdominal masses in dogs is a common finding in clinical practice. While data regarding masses in specific abdominal organs are available in the literature, studies comparing the site distribution of benign versus malignant lesions are scarce. The aim of this study was to retrospectively describe the tissue distribution and diagnosis of surgically excised abdominal masses in a canine population. A total of 123 abdominal masses were classified based on the organ of origin and histologically classified as non-neoplastic (39), benign neoplasia (15), and malignant neoplasia (69). Gastrointestinal masses were more likely to be malignant than masses in other sites. The masses not associated with any organ were significantly larger than the genital and splenic lesions, and no association between size and malignancy was found. This case series suggests that, while the size of the lesion cannot be used as a parameter to predict the probability of a mass being malignant neoplasm, the gastrointestinal site may be used for this scope, providing useful information for primary care clinicians.

**Abstract:**

The detection of an abdominal mass represents a common finding in clinical practice. The aim of this study was to retrospectively describe the tissue distribution and diagnosis of abdominal masses amenable to surgical removal in a canine population. Dogs with abdominal masses with a minimum diameter of 3 cm were selected. Cases were classified, based on the anatomical location, as splenic, gastrointestinal, hepatobiliary, genital, and masses not associated with any organ. Masses were surgically removed and formalin-fixed for the histological examination. Collected data were statistically analyzed. A total of 123 masses were collected from 122 dogs. Sixty-nine masses were classified as malignant neoplasia, 15 as benign, and 39 as non-neoplastic. The abdominal masses were 5.8-fold more likely to be malignant if located in the gastrointestinal tract (*p* = 0.01). A significant association between the size and the site of the masses was identified, the masses not associated with any organ being larger than the genital and splenic lesions (*p* = 0.008). This case series describes the most frequent location in association with the histopathological diagnosis of canine abdominal masses and suggests that the gastrointestinal location was related to a higher risk of representing a malignant neoplasm.

## 1. Introduction

The detection of an abdominal mass represents a common finding in clinical practice, and several differential diagnoses have been recognized in both dogs and cats, including benign [1,2,3,4] and malignant conditions [5,6,7,8]. 

In the majority of the cases, since the clinical presentation is often unspecific, it can be difficult to differentiate the underlying cause and the organ of origin on the basis of history and physical examination alone. 

In human medicine, the majority of abdominal space-occupying lesions are reported to be malignant in origin [9,10,11], and it has been widely accepted that delayed detection and diagnosis can affect the clinical outcome of the patient. 

The treatment and outcome of a patient with an abdominal mass depend on the origin of the mass, the cytological diagnosis (if available), the stage of the disease (in case of malignant lesions), and whether or not surgical treatment is an option. Recently, the frequency of diagnostic imaging procedures increased, and there was a consequent increase in the number of documented masses [12,13,14]. 

In veterinary medicine, epidemiological data regarding canine abdominal masses have been focused on specific diagnoses and sites, with the majority of the studies describing splenic and hepatic masses [12,15,16,17], while data comparing the frequency of benign versus malignant lesions in different sites are lacking. 

Therefore, the aim of this retrospective study was to describe the tissue distribution and diagnosis of abdominal masses amenable to surgical removal in a canine population.

## 2. Materials and Methods

Abdominal masses in dogs sampled for histological diagnosis and referred to a single histopathology diagnostic service were collected from two distinct referral centers between January 2014 and May 2017. Samples were included if they belonged to the canine species, the final histopathological diagnosis was available, and the lesion was at least 3 cm in diameter. Unspecific organ enlargement without a mass, cases for which a histological diagnosis was not conclusive (e.g., necrotic samples), and cases in which the size of the lesion was unknown were excluded. The size of the lesions was measured at the trimming station and the largest diameter was recorded. As the definition of “mass” is not well established and specific guidelines in veterinary medicine are lacking, a cut-off value of 3 cm was established based on human medicine [18] and used as an inclusion criterion. The diagnostic work-up in all patients included a physical examination, a complete blood count, and a serum biochemistry profile. Breed, sex, and age of the affected dogs, as well as the site of the lesion, diameter, and histological diagnosis, were recorded. When performed, the type of imaging that allowed the identification of the mass (thoracic radiographs combined with abdominal ultrasound and/or total body computed tomography) was also recorded. The site of the lesions, based on the distribution identified, was classified as splenic, gastrointestinal, hepatobiliary, genital, and masses not associated with any organ (NAOs). The specific histological diagnoses were recorded and classified as non-neoplastic lesions (non-NPLs), benign neoplasms (B-NPLs), and malignant neoplasms (M-NPLs). Regarding the statistical analysis, in order to compare two groups of lesions with different biological behavior, non-NPLs and B-NPLs were merged as benign lesions (BLs) and compared with malignant lesions (MLs). The statistical analysis was performed using the GraphPad Prism statistical software (GraphPad Software, Inc., v.5, La Jolla, CA, USA). The data were analyzed using the Shapiro–Wilk test for normality. Since the data were not normally distributed, the continuous data were analyzed using the nonparametric Mann–Whitney test to compare two groups and the Kruskal–Wallis test followed by the Dunn’s multiple comparisons test to compare more than two groups. Categorical data were examined using the chi-square test or the Fisher’s exact test, based on the number of groups and the sample size. The association between the malignancy of the lesions (BLs versus MLs) and the mass site was estimated using the odds ratio (OR). A *p*-value ≤ 0.05 was considered significant.

## 3. Results

### 3.1. Clinical Data and Tissue Distribution 

A total of 123 masses were collected from 122 dogs. One dog developed two distinct masses simultaneously and was therefore considered as a single patient. One dog developed two distinct masses, one year apart from the other, and was therefore considered as two distinct patients. Seventy-one dogs were males, 22 of which were castrated; 51 dogs were females, 32 of which were spayed. The male-to-female ratio (M/F) was 1.4. The median age was 11 years (range 1–16 years). There were 50 crossbred dogs (41%); 16 Labrador retrievers (13.1%); 6 golden retrievers (4.9%); and 5 each of beagles, boxers, and German Shepherds (4.1%). Twenty-three other breeds were represented, each by fewer than four dogs.

Diagnostic imaging was performed in 87/123 cases: 18 cases were evaluated by means of an abdominal computed tomography (CT) scan, 64 were evaluated by an abdominal ultrasound, 4 were evaluated by an abdominal ultrasound combined with CT scan, and 1 was evaluated by an abdominal radiograph. The type of diagnostic imaging used and the site of the masses recorded in each case are reported in Table 1.

The median size of the lesions was 7 cm (range 3–30 cm, Figure 1). Seventy-nine masses were splenic, 15 were gastrointestinal, 12 were NAOs, 11 were hepatobiliary, and 6 were genital. Sixty-nine masses were M-NPLs, 39 were Non-NPLs, and 15 were B-NPLs; therefore, there were 69 MLs and 54 BLs (Figure 2). The specific diagnoses are summarized in Table 2.

#### 3.1.1. Splenic Masses

Of the 79 splenic masses, 43 were MLs (54%) and 36 were BLs (46%). MLs included hemangiosarcoma (Figure 3), histiocytic sarcomas, nonangiogenic nonlymphocytic sarcomas (NANLSs), lymphomas, and osteosarcoma. B-NPLs included myelolipomas and hemangiomas. Non-NPLs were represented by hematoma associated with nodular hyperplasia and/or extramedullary hematopoiesis and abscess (Table 2). The median size of the splenic masses was 7 cm (range 3–30 cm). 

#### 3.1.2. Gastrointestinal Masses

Of the 15 gastrointestinal masses, 13 were MLs (87%) and 2 were BLs (13%). MLs included leiomyosarcomas, adenocarcinomas, gastrointestinal stromal tumors (GISTs), lymphomas, hemangiosarcoma, and sarcoma not otherwise specified (NOS). Non-NPLs were rare and represented by steatitis and hematoma. No B-NPLs were recorded. Eleven cases were located in the small intestine, 2 were located in the stomach, and 2 were located in the colon (Table 2). The median size of the gastrointestinal masses was 7.2 cm (range 3.5–15 cm).

#### 3.1.3. Masses Not Associated with Any Organ (NAOs)

Twelve masses were not associated with any organ (NAOs). Of these, seven were MLs (58%) and five were BLs (42%). MLs included sarcomas NOS (Figure 4), neuroendocrine neoplasia, GIST not associated with the intestinal wall, metastatic Sertoli cell tumor, and malignant neoplasia NOS. One B-NPL and four Non-NPLs were also identified (Table 2). The median size of the NAO masses was 10 cm (range 7–30 cm).

#### 3.1.4. Hepatobiliary Masses

Of the 11 hepatobiliary masses, six were MLs (55%) and five were BLs (45%). MLs were hepatocellular carcinomas, and hemangiosarcoma. B-NPLs were hepatocellular adenomas. Non-NPLs were all represented by biliary mucoceles (Table 2). The median size of the hepatobiliary masses was 8.8 cm (range 4–15 cm).

#### 3.1.5. Genital Masses

Of the six genital masses, one was an ML (17%) and five were BLs (83%). The ML was a uterine leiomyosarcoma. The B-NPLs were granulosa cell tumors and seminoma. Non-NPLs included ovarian abscess and uterine hematoma (Table 2). The median size of the genital masses was 4.5 cm (range 3–12 cm).

#### 3.1.6. Statistical Analysis 

The abdominal masses were 5.8-fold more likely to be malignant if located in the gastrointestinal tract compared to other sites (*p* = 0.0133 Fisher’s exact test). The other evaluated sites did not reveal an association with malignancy. The OR of each site in association with malignancy and the corresponding *p* values are specified in Table 3. A statistically significant association between the size and the site of the masses was identified, with the NAO masses being significantly larger than genital and splenic lesions (*p* = 0.0084 Kruskal–Wallis test). Size was not associated with malignancy, nor with sex or breed (*p* = 0.3061 Mann–Whitney test; *p* = 0.6473, *p* = 0.5866 Kruskal–Wallis test). In order to minimize a possible bias regarding size, which could be overestimated by the presence of hematomas in splenic masses, the association between size and malignancy was also analyzed excluding the splenic masses. Nevertheless, an association between the size and the malignancy could not be determined (*p* = 0.3498 Mann–Whitney test). Moreover, an association between malignancy and age, sex, and breed could not be determined (*p* = 0.6618 Mann–Whitney test, *p* = 0.5373; chi-square, *p* = 0.2256 chi-square). The results of the statistical analysis are reported in Table 3.

## 4. Discussion

This study describes the diagnosis and tissue distribution of abdominal masses in a canine population and compares the biological class of the lesions at different sites. When considering the entire study population, the number of malignant lesions was slightly higher than the benign lesions (56% and 44%, respectively). This difference was greatest in the gastrointestinal masses, in which a higher frequency of malignant lesions was found compared to other sites. Interestingly, the majority of the gastrointestinal masses were malignant spindle cell tumors (8 out of 15). This finding was surprising since, in the gastrointestinal tract of the dog, the most frequently reported tumors are adenocarcinomas and lymphomas [19]. This discrepancy could be related to the gross presentation of adenocarcinomas, usually presenting as an annular thickening, thus without forming a mass, or as an intraluminal mass, causing obstruction or sub-obstruction before reaching the size set as an inclusion criterion for this study. Consequently, adenocarcinomas of the gastrointestinal tract were probably excluded due to their different clinical presentation. Similarly, gastrointestinal lymphomas frequently occur as a diffuse thickening of the intestinal wall [20,21]. Furthermore, even when presenting as a mass, lymphomas exfoliate more easily than mesenchymal neoplasia, and a cytological diagnosis may be likely obtained, allowing the tumor to be treated by a medical rather than surgical approach. Thus, the number of lymphomas might have been underestimated in our caseload as intestinal lymphomas, presenting as a mass but not being excised, might have been excluded.

As expected, the splenic masses were the most numerous, and BLs were more frequent than in other sites. Despite the difference not being statistically significant, these data were interesting, since the identification of the splenic origin of an abdominal mass may indicate a higher probability of being a benign entity. The most common diagnosis in this group was consistent with a hematoma, usually associated with nodular lymphoid and hematopoietic hyperplasia. As expected, the most common splenic ML was hemangiosarcoma, while lymphoma was rare in this caseload. The small number of splenic lymphomas may be a consequence, as occurred in the gastrointestinal tract, of the inclusion criteria: diffusely enlarged organs were excluded from the study and only a small number of splenic lymphomas were treated surgically. 

The third-largest group included masses NAO. This group was heterogeneous and included a wide range of conditions, varying from malignant neoplasms to inflammatory diseases without a specific prevalence; therefore, the probability of a lesion being benign rather than malignant seemed less predictable. Mesenchymal neoplasms seemed to prevail and included sarcomas NOS, GISTs, and lipomas; however, this result requires additional validation considering the number of total cases enrolled and the fragmentation of this group.

Hepatic masses were less frequent, and the most common neoplasm diagnosed was hepatocellular carcinoma, paralleling the published data [22,23]. Masses arising from the genital tract were rare since, according to the selection criteria, retroperitoneal masses were excluded, and the low number of genital masses did not allow additional consideration.

The size of the mass, which is often considered an indicator of the biological behavior for neoplastic lesions, was not associated with the final diagnosis in this survey. This result may have been biased by the inclusion of splenic lesions, because hematomas could reach a larger size when compared to other locations. Nevertheless, even removing this variable from the analyzed data, no statistical association could be found, further suggesting that an assumption of malignancy should not be based solely on the size of the mass.

Furthermore, since benign neoplasms are usually smaller than malignant ones, the exclusion of masses smaller than 3 cm might have led to the exclusion of benign lesions and be the reason for lacking a prognostic impact of the size in our study.

The limitations of this study include the sample size and the number and heterogeneity of cases included in each subgroup, its retrospective nature, and the potential bias caused by the submission of cases from referral centers, which might have led to the overestimation of some conditions which were not treated in primary care centers. Due to its retrospective nature, presenting complaints and clinical signs were not available for all patients; therefore, a correlation between histopathology and clinical data was not assessed. 

## 5. Conclusions

This case series describes the most common location and diagnosis of canine abdominal masses, identifying the association of the gastrointestinal location with a higher risk of malignancy. The splenic location was the most frequent and the one with the highest probability, although not statistically significant, of having a benign lesion. The size of the mass was not associated with the diagnosis of malignant neoplasia, and this provides clinical data for future, prospective studies collecting cases from both referral and primary care centers.

## Figures and Tables

**Figure 1 animals-11-00962-f001:**
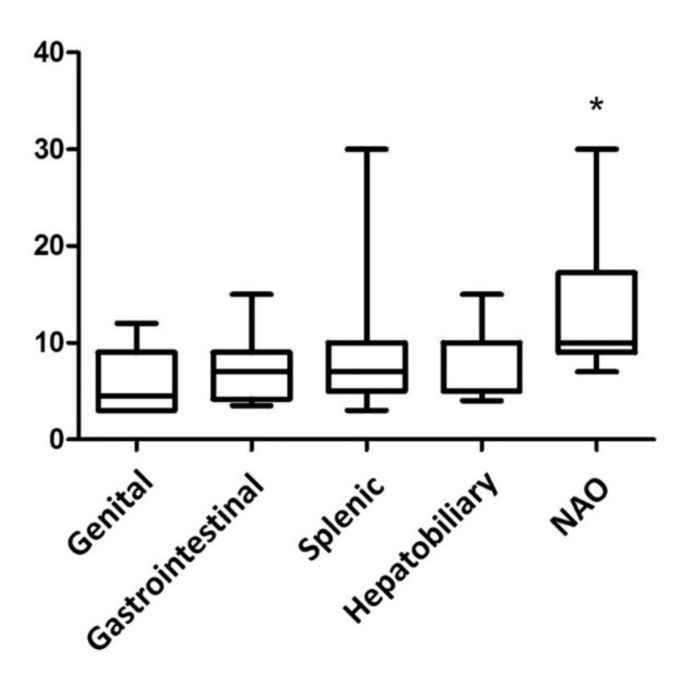
Box plot of the abdominal mass size in various organ sites. The masses not associated with any organs (NAO) are significantly larger than the lesions in the genital tract and in the spleen (*p* = 0.0084 Kruskal–Wallis test and Dunn’s multiple comparison test). * *p*-value ≤ 0.05 considered as significant.

**Figure 2 animals-11-00962-f002:**
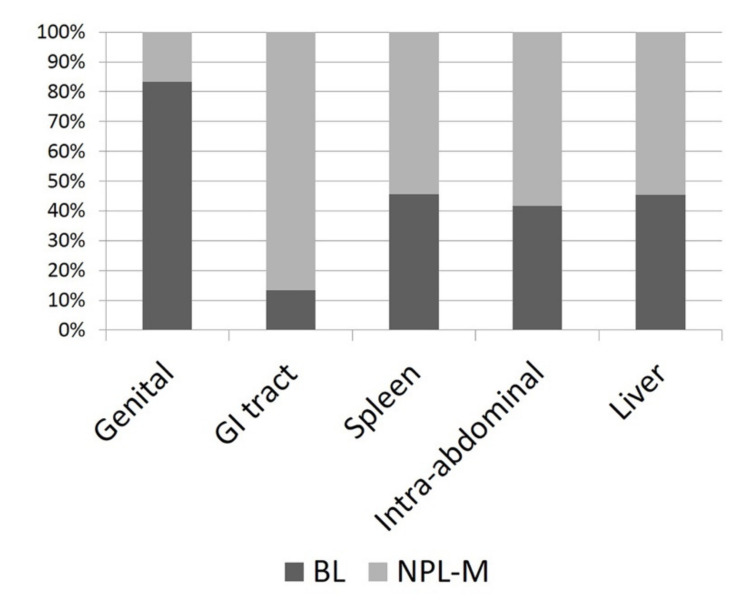
Histogram depicting the percentage of malignant neoplasias (M-NPLs) versus benign lesions (BLs) at different sites of occurrence.

**Figure 3 animals-11-00962-f003:**
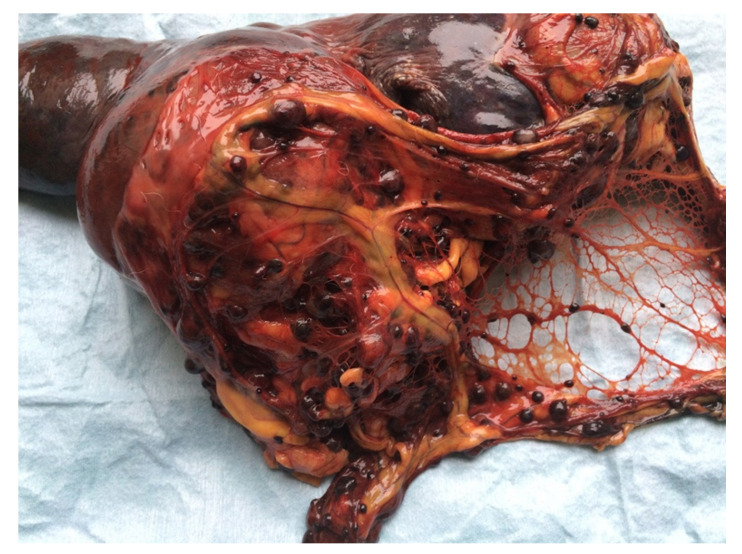
Female neutered crossbred, 13 years old. Splenic hemangiosarcoma, 15 cm in diameter, with multiple metastases on the peritoneum.

**Figure 4 animals-11-00962-f004:**
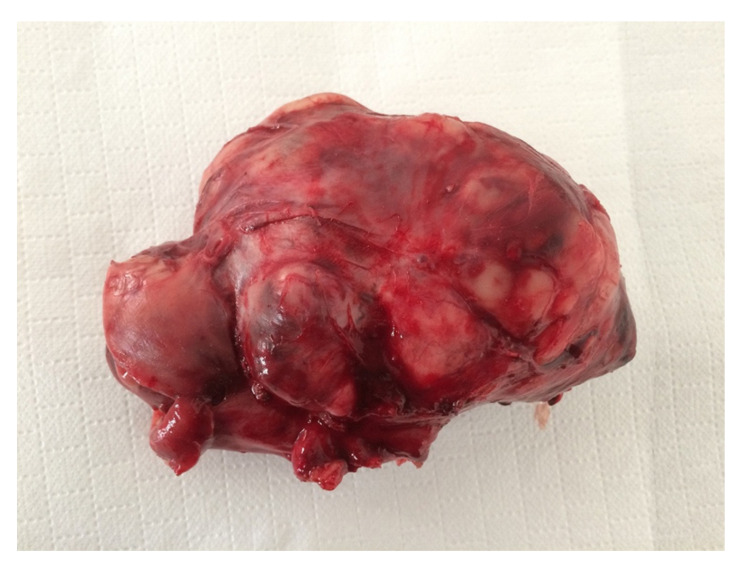
Female neutered crossbred, 14 years old. Intra-abdominal sarcoma, 10 cm in diameter, without connection with any organ.

**Table 1 animals-11-00962-t001:** Type of imaging applied to abdominal masses in dogs. N (%).

Site	CT ^1^	Ultrasound	Ultrasound and CT	Radiography	No Imaging
Splenic	9 (11%)	41 (52%)	3 (4%)	1 (1%)	25 (32%)
Gastrointestinal	2 (13%)	12 (80%)	1 (7%)	0	0
NAO ^2^	7 (58%)	3 (25%)	0	0	2 (17%)
Hepatobiliary	0	5 (45%)	0	0	6 (55%)
Genital	0	3 (50%)	0	0	3 (50%)

^1^ CT: computed tomography, ^2^ NAO: not associated with any organ.

**Table 2 animals-11-00962-t002:** Diagnoses of abdominal lesions in different sites.

Site	Category	Diagnosis	No.
Splenic	ML (M-NPL) ^2^	Hemangiosarcoma	30
		Histiocytic sarcoma	6
		NANLS ^5^	4
		Lymphoma	2
		Osteosarcoma	1
	BL (B-NPL) ^3^	Myelolipoma	6
		Hemangioma	2
	BL (Non-NPL) ^4^	Hematoma	27
		Abscess	1
Gastrointestinal	ML (M-NPL)	Leiomyosarcoma	5
		Adenocarcinoma	2
		GIST ^6^	2
		Lymphoma	2
		Hemangiosarcoma	1
		Sarcoma NOS ^7^	1
	BL (Non-NPL)	Steatitis	1
		Abscess	1
NAO ^1^	ML (M-NPL)	Sarcoma NOS	3
		Neuroendocrine neoplasia	1
		GIST	1
		Sertoli Cell Tumor (metastatic)	1
		Malignant neoplasia NOS	1
	BL (B-NPL)	Lipoma	1
	BL (Non-NPL)	Parasitic granuloma	1
		Hematoma	1
		Abscess	1
		Foreign body reaction	1
Hepatobiliary	ML (M-NPL)	Hepatocellular carcinoma	4
		Hemangiosarcoma	2
	BL (B-NPL)	Hepatocellular adenoma	2
	BL (Non-NPL)	Biliary mucocele	3
Genital	ML (M-NPL)	Leiomyosarcoma	1
	BL (B-NPL)	Granulosa cell tumor	2
		Seminoma	1
	BL (Non-NPL)	Abscess	1
		Hematoma	1

^1^ NAO: not associated with any organ; ^2^ ML: malignant lesion, M-NPL: malignant neoplasia; ^3^ BL: benign lesion, B-NPL: benign neoplasia; ^4^ BL: benign lesion, Non-NPL: non-neoplastic lesion; ^5^ NANLS: nonangiogenic nonlymphogenic sarcoma; ^6^ GIST: gastrointestinal stromal tumor; ^7^ NOS: not otherwise specified.

**Table 3 animals-11-00962-t003:** Odds ratio of malignancy of abdominal lesions in different sites.

Site	Odds Ratio	*p*-Value
Splenic	0.75	0.56
Gastrointestinal	5.81	0.01 *
NAO ^1^	1.06	1.00
Hepatobiliary	0.90	1.00
Genital	0.13	0.08

^1^ NAO: not associated with any organ * *p*-value ≤ 0.05 considered as significant.

## Data Availability

The data presented in this study are available on request from the corresponding author. The data are not publicly available due to privacy.

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
