# Peer review of "The Site of Origin of Canine Abdominal Masses Correlates with the Risk of Malignancy: Retrospective Study of 123 Cases"

_animals, 2021, doi:10.3390/ani11040962_

Round 1
Reviewer 1 Report
The paper is well formulated and well written, the contributions in terms of data and statistics are correct. The photographic contributions are also very comprehensive. Perhaps some microscopic or immunohistochemical images of the most challenging cases in diagnostic terms are missing. In my opinion the work can be accepted in present form.Author Response
We thank the reviewer for the appreciation of the paper.
Reviewer 2 Report
This is a nice and well written study comparing the frequency of benign and malignant lesions in different sites. However, I have some smallish issues. which should be addressed. Most of all, the numbers of different masses are low. This should be emphasised in the discussion.
L61: I think that this is not an epidemiological study.
L68: species
L117-120: This should be moved to text.
L138: Please add the size of the mass in the figure text.
Results: Remove the detailed numbers of different masses from the text as you have them also in Table 2.
L188: entire study population
L221: small number?
L239: The main limitation of the study is probably the small number of cases. With larger number of cases you might have (or not) achieved more statistical significances.
Author Response
Dear Reviewer,
thanks for revising our manuscript. We answer below point by point to your comments.
Reviewer: L61: I think that this is not an epidemiological study.
Authors: We agree with the reviewer this is a descriptive rather than an epidemiologic study, the term epidemiology has been removed from the manuscript
Reviewer L68: species
Authors: Corrected as requested
Reviewer L117-120: This should be moved to text.
Authors: These results are already reported in the text, section 3.1.6. Statistical analysis
Reviewer L138: Please add the size of the mass in the figure text.
Authors: Added as requested
Reviewer Results: Remove the detailed numbers of different masses from the text as you have them also in Table 2.
Authors: Numbers that were also reported in table 2 have been removed from the text. Numbers present only in the text have been maintained.
Reviewer L188: entire study population
Authors: Corrected as requested
Reviewer L221: small number?
L239: The main limitation of the study is probably the small number of cases. With larger number of cases you might have (or not) achieved more statistical significances.
Authors: This limitation has been added in the discussion as requested
Reviewer 3 Report
Thank you for the opportunity to review this study. The authors describe the possible relationship between the site of origin of canine abdominal masses and the risk that they are malignant lesions. They carried out the study with 123 tumors with a size greater than 3 cm. Based on results they suggest that the gastrointestinal location was related to a higher risk of representing a malignant neoplasm. Although the study provides data which could be interesting however, I consider that the information provided has limitations to apply in the clinic practice. Besides there are some mistakes in the paper that must be corrected in the case that it is accepted.
Suggestions/comments:
Introduction sections
I do not understand the criteria to present the references, they start with number 12 and the references 2, 4, 5, 17 and 18 are missing.
Material sections
They should indicate why they do not collect tumors smaller than 3 cm, and indicate this as a limitation of the study.
Results section
In each subsection the authors repeat data which are present in the table 2. I think that they have to eliminate these data from the text, pointing out only the most important data in the text.
On the other hand, the number of tumors they use does not match the total of the tumors in the table 2. They indicate that one animal had two tumors but they considered it as one and another presented two but separated along time and considered it as two different, perhaps they should explain a little better or correct the error, if it exists.
In line 135 they refer to the size of the splenic samples and it is observed that it coincides with the mean size of all the masses, is that correct?
Minor suggestions/comments:
Discussion section
Line 194. It includes an author instead of a reference number, which on the other hand does not appear in the references.
Author Response
Dear Reviewer,
Thank you for revising our manuscript. We answer below point by point to your comments.
Reviewer:
Introduction sections
I do not understand the criteria to present the references, they start with number 12 and the references 2, 4, 5, 17 and 18 are missing.
Authors: We corrected the references in the text. There was an error due to the reference’s manager program.
Reviewer:
Material sections
They should indicate why they do not collect tumors smaller than 3 cm, and indicate this as a limitation of the study.
Authors: The rationale for selecting cases larger than 3 cm is reported in the discussion, and is based on the human literature since, in veterinary medicine, a specific cutoff useful for the definition of a “mass” is lacking. In order to clarify this point, this selection criterion has been also detailed in the materials and methods
Reviewer:
Results section
In each subsection the authors repeat data which are present in the table 2. I think that they have to eliminate these data from the text, pointing out only the most important data in the text.
Authors: This point has been suggested also by reviewers 2 and the repeated data have been removed from the text
Reviewer: On the other hand, the number of tumors they use does not match the total of the tumors in the table 2.
Authors: We thank the reviewer for spotting this discrepancy, in the table we erroneously counted 28 splenic hematomas rather than 27. This has been corrected.
Reviewer: They indicate that one animal had two tumors but they considered it as one and another presented two but separated along time and considered it as two different, perhaps they should explain a little better or correct the error, if it exists.
Authors: The reason for this choice is that in the first case the dog developed simultaneously two masses, thus the impact of breed, age and sex on the risk of development of lesions is the same for both of them. In the second case the dog developed two masses in two distinct moments (more than 1 year of difference) and were therefore considered two separates events for which breed, age and sex may have a different impact.
Reviewer: In line 135 they refer to the size of the splenic samples and it is observed that it coincides with the mean size of all the masses, is that correct?
Authors: Yes, it is correct.
Reviewer:
Minor suggestions/comments:
Discussion section
Line 194. It includes an author instead of a reference number, which on the other hand does not appear in the reference
Authors: We added the reference (n 19) in the references list and we corrected in the text.
Round 2
Reviewer 3 Report
The authors have adequately answered all the doubts / corrections raised. and the paper has improved.
Minimal corrections
Results section
L118 write the first letter of the word table with capital letter.
I think that the authors could do reference to Table 2 in each paragraph: L141, L150, L157, L165, L170.
Author Response
Dear Reviewer,
Thank you very much to further revised our manuscript.
Reviewer: L118 write the first letter of the word table with capital letter.
Authors: Done.
Reviewer: I think that the authors could do reference to Table 2 in each paragraph: L141, L150, L157, L165, L170.
Authors: Done.